# New Insight into the Chemical Composition, Antimicrobial and Synergistic Effects of the Moroccan Endemic *Thymus atlanticus* (Ball) Roussine Essential Oil in Combination with Conventional Antibiotics

**DOI:** 10.3390/molecules26195850

**Published:** 2021-09-27

**Authors:** Ahmed Nafis, Marcello Iriti, Lahcen Ouchari, Fatima El Otmani, Najat Marraiki, Abdallah M. Elgorban, Asad Syed, Noureddine Mezrioui, Lahcen Hassani, Luísa Custódio

**Affiliations:** 1Microbiology, Health and Environment Team, Faculty of Sciences, Chouaïb Doukkali University, El Jadida 24000, Morocco; elotmanifatima@yahoo.fr; 2Laboratory of Microbial Biotechnologies, Agrosciences and Environment, Faculty of Sciences Semlalia, Cadi Ayyad University, Marrakech 40000, Morocco; mezrioui@uca.ac.ma (N.M.); lhassani@uca.ac.ma (L.H.); 3Department of Agricultural and Environmental Sciences, Milan State University, via G. Celoria 2, 20133 Milan, Italy; marcello.iriti@unimi.it; 4Moroccan Coordinated Collection of Microorganisms (CCMM), National Center for Scientific and Technical Research (CNRST), P.O. Box 8027, Rabat 10102, Morocco; ouchari@cnrst.ma; 5Department of Botany and Microbiology, College of Science, King Saud University, P.O. Box 2455, Riyadh 11451, Saudi Arabia; najat@ksu.edu.sa (N.M.); aelgorban@ksu.edu.sa (A.M.E.); assyed@ksu.edu.sa (A.S.); 6Center of Marine Sciences, Faculty of Sciences and Technology, University of Algarve, Ed. 7, Campus of Gambelas, 8005-139 Faro, Portugal; lcustodio@ualg.pt

**Keywords:** thyme, Moroccan endemic plant, essential oil, antimicrobial activity, synergy, antibiotic resistance

## Abstract

This study reported the volatile profile, the antimicrobial activity and the synergistic potential of essential oil (EO) from the Moroccan endemic *Thymus atlanticus* (Ball) Roussine, in combination with the antibiotics ciprofloxacin and fluconazole for the first time, to the best of our knowledge. The EO chemical composition was determined by gas chromatography coupled to mass spectrometry (GC-MS) analysis and the antimicrobial activity assessed by the disc diffusion method against three Gram positive (*Bacillus subtilis*, *Micrococcus luteus*, *Staphylococcus aureus*) and three Gram-negative bacteria (*Pseudomonas aeruginosa*, *Escherichia coli* and one clinical isolate, *Klebsiella pneumonia*). The antifungal activity was evaluated in four pathogenic yeasts (*Candida albicans*, *C. glabrata*, *C. krusei and C. parapsilosis*). The minimum inhibition concentration (MIC) and the synergistic effect with ciprofloxacin and fluconazole were determined by the two-fold dilution technique and checkerboard test, respectively. Twenty-one constituents were identified by GC-MS in the EO, including carvacrol (21.62%) and borneol (21.13%) as the major components. The EO exhibited a significant antimicrobial activity with inhibition zones ranging from 0.7 mm to 22 mm for *P. aeruginosa* and *B. subtilis*, respectively, and MIC values varying from 0.56 mg/mL to 4.47 mg/mL. The fractional inhibitory concentration index (FICI) values ranged from 0.25 to 0.50 for bacteria and from 0.25 to 0.28 for yeasts. The maximum synergistic effect was observed for *K. pneumonia* with a 256-fold gain of antibiotic MIC. Our results have suggested that EO from *T. atlanticus* may be used alone or in association with antibiotics as a new potential alternative to prevent and control the emergence of resistant microbial strains both in the medical field and in the food industry.

## 1. Introduction

Morocco is an important reservoir of biodiversity and Mediterranean speciation with an important flora of around 3913 taxa, including 1298 subspecies in 981 genera and 155 families. This Moroccan plant biodiversity is characterized by a high percentage of endemic species (22%), comprising 878 endemic taxa with 599 taxa at the species level [1].

The genus *Thymus*, belonging to the *Lamiaceae* family, is widely distributed in Morocco with 11 endemic species, including *Thymus atlanticus* (Ball) Roussine, locally known as “Azukni”. This thyme species is characterized by very small white or pale pink flowers with revolute leaves or subplanes, glabrescent and in the form of a condensed carpet. It is the smallest of the Moroccan thymes, and very polymorphic depending on altitude and ecology, growing between 1700 to 3400 m [2]. For a long time, thyme has been commonly used in traditional Moroccan medicine to treat many chronic diseases, especially those related to the digestive system and skin lesions infected with eczema or pathogenic fungi such as *Trichophyton rubrum* [3,4]. Moreover, some studies have reported that a *T. atlanticus* aqueous extract from leaves exhibits relevant anti-inflammatory and antioxidant activities due to its high content in total phenolic compounds [5]. In addition, Khouya et al. [6] showed that the aqueous polyphenol-rich extract of this plant inhibited the blood coagulation ex vivo and in vivo. However, to the best of our knowledge, the antimicrobial activity of *T. atlanticus* has never been evaluated.

Previous reports showed that Moroccan endemic thymes exhibited important biological properties, including antifungal, antimicrobial, antiviral and antioxidant activities [7,8,9]. Furthermore, it was demonstrated that extracts from other Moroccan endemic thymes, such as *T. riatarum*, *T. maroccanus* and *T. broussonetii*, exhibited a high antimicrobial efficacy in combination with standard antibiotics as new efflux pump inhibitors or as membrane permeabilizers [7,10]. In general, the combination of antimicrobial drugs with natural products from medicinal plant characterized by a multicomponent chemical composition and multitarget mechanism of action provides many benefits, such as the decrease of the toxic effects of the combined active ingredients. This new and safe alternative can solve the problem of antimicrobial resistance that has become a serious public health concern with economic, social and medical implications, especially in the case of multidrug-resistant microbes and resulting infections. In this context, this study aimed to evaluate, for the first time, the antimicrobial and antifungal activities of the essential oil obtained from aerial organs of the Moroccan endemic *T. atlanticus*, along with its synergistic effect with standard antimicrobials (ciprofloxacin and fluconazole).

## 2. Results and Discussion

The steam distillation of *T. atlanticus* aerial parts provided a yellow oil with a yield of 1.41%, based on dry plant material (*v*/*w*). The obtained yield was similar to the values reported for some thyme species, although it was different when compared with others. For example, Fadli et al. [7] reported that the yield of the EOs obtained from *T. maroccanus* and *T. broussonetii* were 1.38% and 1.2%, respectively. In another work, the *T. riatarum* yield was 0.26% [10]. Mahboubi et al. [11] obtained yields of Iranian thymes ranging from 2% to 2.2% for *T. vulgaris* and *T. daenensis*, and from 0.68% to 1.1% for *T. kotschyanus*, *T. fedtschenkoi* and *T. pubescens*.

The GC-MS analysis of the *T. atlanticus* EO revealed the presence of 21 volatile compounds (Table 1 and Figure 1) representing 97.44% of total oil. The major components were carvacrol (21.62%) and borneol (21.13%), followed by *γ*-terpinene (9.98%), *o*-cymene (8.14%) and camphene (7.28%). Monoterpenes (88.91%) were the most represented compounds. Quantitative differences in the chemical profile were noted in comparison with Fadli et al., (2014) in which borneol (41.67%) was the most abundant component of the *T. riatarum* EO, followed by *α*-terpineol (8.65%), whereas carvacrol (0.75%) was present in small amounts. Fadli et al. [7] also reported a different composition, with carvacrol (76.35% and 39.77%) and borneol (0.41% and 12.03%) as the main constituents of EO from Moroccan endemic thymes (*T. maroccanus* and *T. broussonetii*, respectively). A recent study revealed differences in the chemical composition of *T. vulgaris* EO, where thymol (55.3%) was identified as the major metabolite [12]. Furthermore, Mahboubi et al. [11] demonstrated that EOs extracted from the Iranian chemotypes *T. fedtschenkoi* and *T. pubescens* were rich in thymol (50.6% and 26.6%, respectively) and carvacrol (6.6% and 27%, respectively). Based on the obtained results, the variation between the chemical composition of our samples and those determined for other thyme species can depend on several factors such as the geographical location, climate, plant material and the season of the material collection [13].

The disk diffusion assay showed that the EO had a broad-spectrum antimicrobial activity, with inhibition zones (IZ) ranging from 7 mm (in *P. aeruginosa*) to 22 mm (in *B. subtilis*) (Table 2). The EO showed a powerful inhibitory activity against *B. subtilis* (IZ = 22 mm) and both Gram-negative bacteria, namely *E. coli* (IZ = 18 mm) and *K. pneumoniae* (IZ = 16 mm) (Table 2). The EO also exhibited a high antifungal potential against the pathogenic yeasts *C. albicans* (IZ = 15 mm), *C. glabrata* (IZ = 18 mm), *C. krusei* (IZ = 20 mm) and *C. parapsilosis* (IZ = 12 mm) (Table 2). The MIC values showed that the EO inhibited all the tested microorganisms at a concentration of 0.56 mg/mL, except for *S. aureus* and *P. aeruginosa*, which were inhibited at 1.12 mg/mL and 4.47 mg/mL, respectively (Table 2). Remarkably, the Gram-negative bacterium *P. aeruginosa* seemed to be the least sensitive. Overall, the EO exhibited a significant antimicrobial activity, probably due to its chemical profile rich in carvacrol and borneol. The oxygenated monoterpene cavacrol is a well-known and efficient antimicrobial agent that can damage the cell membrane or other cytoplasmic targets passing through the altered phospholipid bilayer [14,15,16]. Similarly, the bicyclic monoterpene borneol is also a powerful and broad-spectrum antimicrobial agent able to alter the membrane structural and functional integrity [17]. In addition, other volatile compounds present in lower percentages can also contribute to the biological activity of the EO, including *γ*-terpinene, linalool and thymol [18,19,20]. Concerning the resistance of *P. aeruginosa*, it is well known that a multidrug-resistant bacterium uses many different mechanisms to counteract the antimicrobial drugs [21]. In general, the major mechanisms that *P. aeruginosa* has developed were the lower outer membrane permeability, as intrinsic resistance, and the biofilm formation, as adaptive resistance, that serves as a diffusion barrier to limit the access of bioactive substances into the bacterial cells [22].

As reported in Table 3 and Table 4, a total synergism was observed in the 10 tested combinations of *T. atlanticus* EO and both antibiotics (ciprofloxacin and fluconazole). The best synergistic effect was recorded in *K. pneumoniae* and *C. parapsilosis*, with FICI values of 0.25. The gain of antibiotic MIC in the presence of EO ranged from 4- to 256-fold. The highest reduction of fluconazole MIC was observed against *C. parapsilosis* (256-fold), whereas the three other yeast strains had a 32-fold gain. For Gram-positive bacteria, a high decrease was recorded for *K. pneumoniae*, *E. coli* and *M. luteus*, from 256- to 64-fold. Our results showed that the *T. atlanticus* EO enhanced the antimicrobial activity of the reference antibiotic drugs. Previous studies reported the interaction between antimicrobial agents and other Moroccan thyme EOs. For example, Fadli et al. [10] demonstrated that the addition of *T. riatarum* EO against *E. coli* AG100 reduced the chloramphenicol MIC from 8 mg/mL to 2 mg/mL. Similarly, Saad et al. [9] showed that the synergistic effect of *T. maroccanus* and *T. broussonetii* EOs, in combination with amphotericin B and fluconazole, resulted in FICI values of 0.49, 0.27, 0.37 and 0.3, respectively, and decreased the antibiotic MICs with more than an 8-fold gain.

Our results contribute to substantiate the traditional uses of *T. atlanticus* by the Moroccan populations for the treatment of skin infections caused by different types of microorganisms, thus suggesting that its EO could be further investigated as a natural source of adjuvant agents to be used in combination with conventional antibiotics to reduce the risk of selecting antibiotic resistant microbial strains.

## 3. Materials and Methods

### 3.1. Plant Material and Essential Oil (EO) Distillation

The aerial parts of *T. atlanticus* were collected by M. Brahim Oummad in September 2019 from the Imider municipality in the Tinghir province of the Drâa-Tafilalet administrative region of Morocco (31.352509° N, 5.832241° W). Samples were placed at 25 °C in shade to air-dry for one week and their voucher specimens (THAL-46) were conserved at the laboratory of Microbial Biotechnologies, Agrosciences and Environment, Faculty of Sciences Semlalia, Cadi Ayyad University, Marrakech, Morocco. The identification of this thyme species was carried out by Nafis Ahmed, based on the dichotomous keys of Moroccan flora. Dried leaves were subjected to steam distillation for 3 h and the preparation of the essential oil was performed three times (3 × 50 g). Then, the recovered EO was stored at 4 °C in darkness. The yield percentage was calculated as volume (in mL) of EO by 100 g of dried plant material. 

### 3.2. Gas Chromatography-Mass Spectrometry (GC-MS) Analysis

The GC-MS system was used to qualitatively and quantitatively characterize the chemical composition of *T. atlanticus* EO, as previously described [23]. Gas chromatographic coupled to mass spectrometric (GC-MS) analysis was performed on a 1300 GAZ gas chromatograph equipped with a TG-5MS column (30 m length; 0.25 mm i.d.; 0.25 μm film thickness) and coupled to mass selective detector “ISQ Single Quadrupole Mass spectrometer” (70 eV). The analytical conditions were: carrier gas, helium; injection volume was 1 μL; injector temperature 260 °C, temperature program was 1 min at 100 °C, ramped from 100 °C to 260 °C at 4 °C/min and 10 min at 246 °C. The individual identification of each volatile compound was based on the comparison of the obtained mass spectra with NIST and Wiley library reference data and the Kovats retention indices (RI) with a reference library of a series of C9 to C24 *n*-alkanes.

### 3.3. Microorganisms and Culture Conditions

The antibacterial activity of the EO was evaluated against three Gram-positive bacteria, namely *Bacillus subtilis* (ATCC 9524), *Micrococcus luteus* (ATCC 10240), *Staphylococcus aureus* (CCMM B3) and three Gram-negative bacteria, i.e., *Pseudomonas aeruginosa* (DSM 50090), *Escherichia coli* (ATCC 8739) and clinically isolated *Klebsiella pneumoniae*. In addition, the antifungal activity was assessed using the following four pathogenic yeasts: *Candida albicans* (CCMM-L4), *C. glabrata* (CCMM-L7), *C. krusei* (CCMM-L10) and *C. parapsilosis* (CCMM-L18) [24]. The bacterial and yeast species were cultivated for previous use on Muller Hinton agar (at 37 °C) and Sabouraud dextrose agar (at 28–30 °C) plates, respectively.

### 3.4. Minimum Inhibitory Concentration (MIC) and Synergistic Effect Determination

MIC was determined using the two-fold dilution method, as previously described [25]. Briefly, Each microwell was contained 100 μL of oil dilution and 100 μL of cell suspension prepared by dilution (1/100) an overnight culture in MHB and Sabouraud media for bacteria (10^6^ CFU/mL) and yeasts (1–2 × 10^3^ cells/mL), respectively. The MIC value corresponded to the well with the lowest concentration of EO and without any visible microbial growth. The ciprofloxacin and fluconazol MIC values were also determined.

The synergistic effect between the EO and ciprofloxacin and fluconazole was evaluated by determining the antibiotic MIC in presence of the EO at a sub-inhibitory concentration (MIC/4), and the antibiotic MICs were performed using the checkerboard method as previously reported [26]. Briefly, 50 µL of the EO (at MIC/4) and each serial two-fold dilution of antibiotics were inoculated with 100 µL of a cell suspension of each strain (approximately 10^6^ UFC/mL). MIC values were defined as the lowest concentration of antibiotics, in combination with the EO at MIC/4, inhibiting the visible growth of tested microorganisms. All the tests were performed in triplicate.

To determine the combined effect of antimicrobials and the EO, we calculated the fractional inhibitory concentration index (FICI) which is the most frequently used to define or to describe drug interactions. FICI values were defined according to Odds, 2003 [27]: no interaction (FICI between 0.5 and 4), synergy (FICI ≤ 0.5) and antagonism (FICI ≥ 4). The MIC gain of antibiotics was calculated as the antibiotic MIC alone divided by the MIC of antibiotic combined with the EO.

## 4. Conclusions

The *T. atlanticus* EO was characterized by 21 volatile constituents, and carvacrol (21.62%) and borneol (21.13%) were found to be the most abundant. The EO exhibited a significant antimicrobial activity, with MIC values varying from 0.56 mg/mL to 4.47 mg/mL. Moreover, EO showed a total synergistic effect with both antibiotics towards all tested strains and the highest gain in term of antibiotic MIC value was reported against *K. pneumonia*, with a 256-fold gain. Our results suggested that the EO extracted from *T. atlanticus* should be further explored with the aim of investigating its potential use as a tool for the management of multi-drug resistant microorganisms.

## Figures and Tables

**Figure 1 molecules-26-05850-f001:**
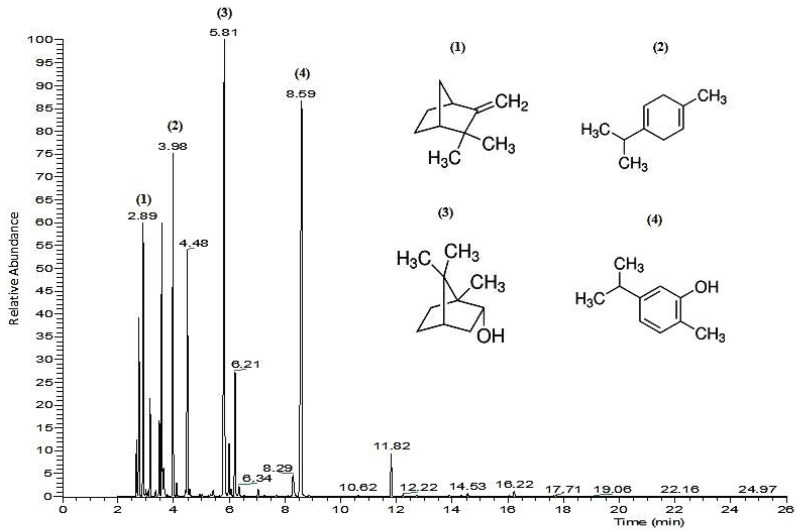
GC-MS chromatogram of *T. atlanticus* essential oil. (**1**): Camphene; (**2**): γ-Terpinene; (**3**): Endo-borneol; (**4**): Carvacrol.

**Table 1 molecules-26-05850-t001:** Components of essential oil distilled from the leaves of Moroccan endemic *T. atlanticus*. Entries in **bold** are major constituents.

RI	Abundance %	Compounds
841	1.55	(E)-2-Hexenal
865	4.4	3-Heptanone
953	**7.28**	Camphene
978	0.16	1-Octen-3-ol
991	2.77	*α*-Myrcene
1088	1.93	Terpinolene
1020	**8.14**	*o*-Cymene
1062	**9.98**	*γ*-Terpinene
1026	0.43	cis-Sabinene hydrate
1098	7.8	Linalool
1078	0.23	cis-4-Thujanol
1143	0.22	Camphor
1165	**21.13**	endo-Borneol
1177	1.69	L-4-terpineol
1189	4.26	*α*-Terpineol
1242	0.34	Carvone
1290	1.31	Thymol
1298	**21.62**	Carvacrol
1454	1.81	Caryophyllene
1526	0.15	*δ*-Cadinene
1581	0.24	Caryophyllene oxide
	88.91	Monoterpenes
	2.2	Sesqueterpenes
	6.33	Others
	97.44	Total

RI: Retention index measured relative to *n*-alkanes (C-9 to C-24) on a non-polar TG-5MS column.

**Table 2 molecules-26-05850-t002:** Inhibition zone (IZ) diameters and minimum inhibitory concentrations (MIC) of the essential oil (EO) extracted from the leaves of Moroccan endemic *Thymus. atlanticus* against bacteria and yeasts using the disc diffusion and micro-well dilution assays.

Microorganisms	EO		Ciprofloxacin		Fluconazole	
	IZ (mm)	MIC (mg/mL)	IZ (mm)	MIC (mg/mL)	IZ (mm)	MIC (mg/mL)
Gram-positive bacteria						
*Staphylococcus aureus*	13.0	1.12	26.0	0.01	-	-
*Micrococcus luteus*	12.0	0.56	27.0	0.03	-	-
*Bacillus subtilis*	22.0	0.56	35.0	0.01	-	-
Gram-negative bacteria						
*Escherichia coli*	18.0	0.56	12.0	0.06	-	-
*Klebsiella pneumoniae*	16.0	0.56	09.0	01.0	-	-
*Pseudomonas aeruginosa*	07.0	4.47	08.0	0.25	-	-
Yeasts						
*Candida albicans*	15.0	0.56	-	-	20.0	1
*Candida glabrata*	18.0	0.56	-	-	13.0	1
*Candida krusei*	20.0	0.56	-	-	24.0	1
*Candida parapsilosis*	12.0	0.56	-	-	28.2	1

IZ: inhibition zones (mm). MIC: minimum inhibitory concentration (mg/mL).

**Table 3 molecules-26-05850-t003:** Synergistic interaction between *T. atlanticus* essential oil (EO) and ciprofloxacin against resistant bacteria.

	*Micrococcus luteus*			*Staphylococcus aureus*			*Bacillus subtilis*			*Escherichia coli*			*Pseudomonas aeruginosa*			*Klebsiella pneumoniae*		
	FIC	FICI	Gain	FIC	FICI	Gain	FIC	FICI	Gain	FIC	FICI	Gain	FIC	FICI	Gain	FIC	FICI	Gain
EO	0.25	-	-	0.25	-	-	0.25	-	-	0.25	-	-	0.25	-	-	0.25	-	-
Ciprofloxacin	0.02	0.27 ^a^	64	0.25	0.50 ^a^	4	0.06	0.31 ^a^	16	0.02	0.27 ^a^	64	0.25	0.50 ^a^	4	0.00	0.25 ^a^	256

FIC: fractional inhibitory concentration. FICI: fractional inhibitor concentration index. FIC of oil = MIC of EO in combination with antibiotic/MIC of EO alone. FIC of antibiotic = MIC of antibiotic in combination with EO/MIC of antibiotic alone. FIC index = FIC of EO + FIC of antibiotic. ^a^ Total synergism.

**Table 4 molecules-26-05850-t004:** Synergistic interaction between *T. atlanticus* essential oil (EO) and fluconazole against clinical pathogenic yeasts.

	*Candida albicans*			*Candida glabrata*			*Candida krusei*			*Candida parapsilosis*		
	FIC	FICI	Gain	FIC	FICI	Gain	FIC	FICI	Gain	FIC	FICI	Gain
EO	0.25	-	-	0.25	-	-	0.25	-	-	0.25	-	-
Fluconazole	0.03	0.28 ^a^	32	0.03	0.28 ^a^	32	0.03	0.28 ^a^	32	0.00	0.25 ^a^	256

FIC: fractional inhibitory concentration. FICI: fractional inhibitor concentration index. FIC of oil = MIC of EO in combination with antibiotic/MIC of EO alone. FIC of antibiotic = MIC of antibiotic in combination with EO/MIC of antibiotic alone. FIC index = FIC of EO + FIC of antibiotic. ^a^ Total synergism.

## Data Availability

The data presented in this study are available in the article.

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
