# Peer review of "New Insight into the Chemical Composition, Antimicrobial and Synergistic Effects of the Moroccan Endemic Thymus atlanticus (Ball) Roussine Essential Oil in Combination with Conventional Antibiotics"

_molecules, 2021, doi:10.3390/molecules26195850_

Round 1

Reviewer 1 Report

The submitted manuscript is generally well written and the experimental outcomes are properly reported and discussed. However, I have found some missing details that I want to report in the following and bring to the attention of authors:

1) pag. 2, yield 1.41%. It is not clear if this is on a weight basis.

2) pag. 3. authors assert in the manuscript that the volatile compounds are 97.44% of the total oil. How this was extimated/calculated?

3)  pag. 5. authors should make more explicit how FIC, FICI and Gain were calculated and what do they mean. I think that reader needs to better understand why these parameters or indexes have been used.

4) section 3.2, pag. 6. authors should briefly report the operational parameters employed for the GC separation (i.e., temperatures, column, carrier pressure, etc.) and also type of MS analyzer and the library used for compound identification from MS spectra

Author Response

Response comments to the Editorial Office of “Frontiers in Microbiology” and reviewers.

The expert comments were really found very positive and constructive. We have now thoroughly revised the manuscript in response to reviewer kind and healthy suggestions.

In this letter, we have provided a detailed response and changes made in the light of the reviewer comments.

Reviewer 1

The submitted manuscript is generally well written and the experimental outcomes are properly reported and discussed. However, I have found some missing details that I want to report in the following and bring to the attention of authors:

1) pag. 2, yield 1.41%. It is not clear if this is on a weight basis.

- It has been done.

2) pag. 3. authors assert in the manuscript that the volatile compounds are 97.44% of the total oil. How this was extimated/calculated?

- We calculated the percentage of total oil by the sum of the different percentages of each volatile compound detected (see table 1).

3)  pag. 5. authors should make more explicit how FIC, FICI and Gain were calculated and what do they mean. I think that reader needs to better understand why these parameters or indexes have been used.

 - It has been done. The explanation of required information was added in the text and below the tables.

4) section 3.2, pag. 6. authors should briefly report the operational parameters employed for the GC separation (i.e., temperatures, column, carrier pressure, etc.) and also type of MS analyzer and the - library used for compound identification from MS spectra

- It has been done.

Reviewer 2 Report

The topics is really hot.

The subject of the article is extremely important and actual. 

I have the following comments and questions for the authors. There are many awkward phrases that I do not point out here; I only point out those where the meaning cannot be interpreted:

The paragraph 3.4 first paragraph is not clear can be rewrite in more clear format.

Figure 1 is not clear need to be reformat. 

The conclusion need to clear and specific. My recommendation is to focus on short conclusion.

Please recheck the References order.

Please double check the article by a native English reader.

Thanks again for the chance of reading the article.

Author Response

Response comments to the Editorial Office of “Frontiers in Microbiology” and reviewers.

The expert comments were really found very positive and constructive. We have now thoroughly revised the manuscript in response to reviewer kind and healthy suggestions.

In this letter, we have provided a detailed response and changes made in the light of the reviewer comments.

Reviewer 2

The paragraph 3.4 first paragraph is not clear can be rewrite in more clear format.

- It has been done.

Figure 1 is not clear need to be reformat. 

- It has been done

The conclusion need to clear and specific. My recommendation is to focus on short conclusion.

- It has been done.

Please recheck the References order.

- It has been done.

Please double check the article by a native English reader.

- It has been done.

Thanks again for the chance of reading the article.

- Many thanks for your dear reviewer.